# Exploiting the Natural Properties of Extracellular Vesicles in Targeted Delivery towards Specific Cells and Tissues

**DOI:** 10.3390/pharmaceutics12111022

**Published:** 2020-10-26

**Authors:** Pablo Lara, Alan B. Chan, Luis J. Cruz, Andrew F. G. Quest, Marcelo J. Kogan

**Affiliations:** 1Percuros B.V., 2333 CL Leiden, The Netherlands; p.lara_arenas@lumc.nl (P.L.); achan@percuros.com (A.B.C.); 2Translational Nanobiomaterials and Imaging (TNI) Group, Radiology Department, Leiden University Medical Center, Albinusdreef 2, 2333 ZD Leiden, The Netherlands; L.J.Cruz_Ricondo@lumc.nl; 3Advanced Center for Chronic Diseases (ACCDiS), University of Chile, Santos Dumont 964 Independencia, 8380000 Santiago, Chile; 4Laboratory of Cellular Communication, Program of Cell and Molecular Biology, Center for Studies on Exercise, Metabolism and Cancer (CEMC), Institute of Biomedical Sciences (ICBM), Faculty of Medicine, University of Chile, Av. Independencia 1027, 8380453 Santiago, Chile; 5Departamento de Química Farmacológica y Toxicológica, Facultad de Ciencias Químicas y Farmacéuticas, Universidad de Chile, Santos Dumont 964 Independencia, 8380494 Santiago, Chile

**Keywords:** extracellular vesicles (EVs), exosomes, microvesicles, targeting, tracking, drug delivery, theranostics

## Abstract

Extracellular vesicles (EVs) are important mediators of intercellular communication that participate in many physiological/pathological processes. As such, EVs have unique properties related to their origin, which can be exploited for drug delivery applications in cell regeneration, immunosuppression, inflammation, cancer treatment or cardioprotection. Moreover, their cell-like membrane organization facilitates uptake and accumulation in specific tissues and organs, which can be exploited to improve selectivity of cargo delivery. The combination of these properties with the inclusion of drugs or imaging agents can significantly improve therapeutic efficacy and selectivity, reduce the undesirable side effects of drugs or permit earlier diagnosis of diseases. In this review, we will describe the natural properties of EVs isolated from different cell sources and discuss strategies that can be applied to increase the efficacy of targeting drugs or other contents to specific locations. The potential risks associated with the use of EVs will also be addressed.

## 1. Introduction

The ability of drugs to exert their therapeutic effects is limited by their stability in circulation and their capacity to cross cellular barriers and reach the desired tissue. In cancer, for example, most therapies have limited efficacy as drugs have low selectivity, which results in a considerable number of side effects in the organism. For this reason, efforts currently focus on the development of therapeutic agents that can be targeted to specific sites in the body. The availability of such agents would improve the therapeutic opportunities, the efficiency of the treatment and the survival of the patients, while reducing undesired side effects.

The utilization of nanomaterials has revolutionized research in drug delivery due to the physical and chemical characteristics of nanoscale materials. Moreover, nanoparticles (NPs) have the potential to combine multiple therapeutic functions on the same platform, for example by incorporating drugs or agents that increase cell penetration, labelling agents or biopolymers, among others. Current strategies aim to develop intelligent nanomaterials that incorporate multiple functions and are capable of selectively reaching the therapeutic target, diagnosing the disease and carrying out treatment simultaneously.

Despite the great potential of nanomaterials, the majority of synthetic NPs developed never reach clinical trials, because they fail to overcome the multiple barriers present in the organism. Most of the nanoparticles are captured by the mononuclear phagocytic system and retained in the liver and spleen for subsequent elimination. The NPs that manage to overcome these barriers must cross others, such as the blood-brain barrier that prevents the passage of 99% of the molecules. Furthermore, to reach their intended cellular location, NPs are confronted with other obstacles, such as poor vascularization in the case of cancer cells, cell impermeability, endosomal escape, as well as resistance mechanisms involving efflux pumps [1].

As a result, there is a need for nanovehicles with the ability to evade these multiple barriers in the organism and at the same time increase selective targeting to specific cellular locations. Recently, the utilization of extracellular vesicles (EVs) for drug delivery in different fields of therapeutics has gained popularity as they are natural carriers of biological material between cells [2,3,4,5,6,7,8,9,10,11]. These vesicles are secreted by almost all cell types and can be isolated from different body fluids, such as urine, blood and cerebrospinal fluid, as well as from other external sources, such as plants, fruits and milk. The EV contents are determined by their origin and include various cell-specific molecules, such as integrins, immunoglobulin family members, heat-shock proteins, RNA, miRNA, antigen-presenting proteins and tetraspanins, which make them interesting for diagnostics and immunotherapy. EVs have also been shown to be highly tunable structures and efficient vehicles for drug delivery [12]. As the homing properties of these vesicles are determined by specific cell-membrane components, the drug selectivity can be improved by isolating EVs with natural tropism to the brain, liver, lung, cancer cells or others. These properties can be further enhanced by loading EVs with drugs, lipids, peptides, NPs, imaging agents or by engineering cells to produce EVs that express a specific molecule to improve their targeting or therapeutic effectiveness [13,14,15,16].

Designing a good strategy for targeted therapy can be challenging when considering the multiple alternatives of EV-producing cells or biological fluids, the different properties of each type of EV and the targeting/drug-loading methods currently available. Choosing the most appropriate strategy depending on the therapeutic target can have a great impact on therapy outcome. In this review, we will focus particularly on the utilization of the natural properties of EVs to favor targeting and efficacy towards specific cells and discuss different strategies to enhance and combine that potential for cell-specific targeting, drug delivery and imaging purposes. Further, the potential risks and limitations in the use of EVs will be discussed.

## 2. Extracellular Vesicles

EVs are particles surrounded by a lipid bilayer which are released by most eukaryotic and prokaryotic cells as a means of intercellular communication in an evolutionarily conserved process [17,18]. EVs can be found in different body fluids, such as blood, saliva, urine, seminal fluid, and breast milk. Importantly, increasing evidence points towards their potential to serve as biomarkers in the diagnosis and prognosis of a variety of pathologies [19,20,21,22,23]. These vesicles are capable of transporting cytosolic and membrane proteins, including receptors and major histocompatibility complexes, as well as DNA fragments and RNA molecules (mRNA, microRNA and other non-coding RNAs) and even organelles (large EVs) [19,24,25,26]. Moreover, EVs display different properties depending on the cell type from which they were isolated. For instance, EVs from immune cells express markers, such as MHC or CD3 molecules, on their surface, which allow them to trigger specific responses in the immune system [19,27].

According to their biogenesis, EVs can be separated into three main groups—exosomes, microvesicles (MVs) and apoptotic bodies. Exosomes are stored in multivesicular bodies (MVBs) and subsequently released into the extracellular space when these bodies merge with the plasma membrane. MVs are formed by shedding from the plasma membrane and apoptotic bodies are released by dying cells. Since there is still no consensus on a specific marker for EV-specific subtypes [17], the international society for extracellular vesicles recommends using a general terminology rather than assigning the EVs to a particular biogenesis pathway. It has also been suggested to segregate them into subcategories according to their size (small, medium or large EVs), their density or their biochemical composition (i.e., CD63+). However, given that there is still no standardized nomenclature for the different vesicle subpopulations, we will use the more general term and refer to them as EVs throughout this review, focusing on summarizing data from publications relating to exosomes and MVs.

### EVs for Drug Delivery

EVs have great potential for drug delivery, due to their natural properties and versatility which we will address in the following sections. These vesicles have an intrinsic capacity to cross biological barriers, are capable of transporting various components and protect their content from degradation [26,28,29]. As natural regulators of the cellular microenvironment, they also play an important role in cellular communication [30,31,32]. EVs are endowed with specific properties related to their biogenesis that can be used to improve the effectiveness of therapy [4,11,33,34,35,36,37]. Examples of this are oral absorption (milk EVs [11]), anti-inflammatory effect (grape EVs [38]), and presence of specific receptors (e.g., transferrin, major histocompatibility complexes, folate [27,39,40,41]). Some EVs can also increase their residence time in circulation by displaying antiphagocytic surface markers to evade clearance by the mononuclear phagocytic system [42]. Different compounds can be incorporated into such vesicles, such as drugs, nanoparticles, lipids, proteins, peptides, RNA, siRNA and fluorescent markers [13,14,15]. Lipophilic drugs or compounds will preferentially intercalate into the membrane bilayer, while hydrophilic compounds will prefer the lumen. Additionally, the surface of the EVs can be modified to improve their functionality. Various techniques have been proposed to favor the inclusion of therapeutic molecules in EVs. One of the most used is electroporation, which involves applying an electric field to a suspension containing the EVs and the active molecule, to create pores that facilitate movement of the drug to the vesicle interior [43]. Electroporation has the advantage of not requiring a vector to incorporate the drug and benefits from the technology developed to include molecules in liposomes. However, it can affect colloidal stability, causing EV properties to be lost or altered. It can also promote aggregation of the loaded molecules such as siRNA and it is overall difficult to scale up [19,44,45]. Some authors have also proposed the possibility of including drugs in EVs by sonication or direct incubation of the vesicles with the active ingredients [6,46]. The sonication method is useful for encapsulating lipophilic and hydrophilic drugs because the vibrations disrupt the membranes and facilitate entry. However, this method is only applicable for small molecules and might cause the adhesion of the molecule to the EV surface, which could change the biodistribution and release of the drug [19,47]. On the other hand, direct incubation is one of the simplest techniques and usually does not require the addition of any additional reagents; however, it is also limited to small and generally lipophilic compounds [19,46]. Another method used is cell transfection that involves transferring specific genes to the parental EV-producing cells where the molecule of interest is generated and then included during EV biogenesis [48,49]. Currently, there are genetic editing tools that facilitate the use of this technique; however, they are usually time consuming and difficult to scale up. Although the cell transfection technique was originally proposed exclusively to encapsulate genetic material inside EVs, some authors have proposed using the same concept to incorporate other types of active ingredients. For example the antineoplastic drug Paclitaxel has been incorporated into stromal mesenchymal cells, which were resistant to the drug and incorporated it into EVs, which displayed cytotoxic effects against cancer cells [50]. Another recent example is the incorporation of gold nanoparticles (AuNPs), which were incubated and taken up by cells to promote their passage through the MVB and inclusion in EVs [13]. The pre-cargo-loading strategies permit obtaining EVs containing therapeutic compounds without the need for membrane-disruptive techniques. Further, such EVs can then be modified post-isolation with lipophilic drugs, fluorescent markers or targeting agents (Figure 1).

## 3. Relevant Parameters Involved in the Application of EVs for Drug Delivery

### 3.1. Targeting Properties

As mentioned above, EVs can be loaded with multiple therapeutic agents simultaneously (e.g., targeting peptides, drugs and imaging agents) which makes them highly versatile vectors for drug delivery strategies. However, what makes EVs even more interesting in comparison to other drug delivery vehicles is the variety of natural properties they possess. Although their genesis is still not well understood, available evidence suggests that they have multiple therapeutic benefits for targeting strategies, such as enhanced cellular uptake, organ tropism and immunomodulation. These properties are endowed by the presence of different adhesion and immunoregulatory molecules, as well as cell-specific receptors, which can be used to enhance accumulation in specific tissues. Examples of the latter are shown in Figure 2 and Table 1. In the next section, we will focus on these natural properties and discuss how they can be finetuned to specific applications by isolating the EVs from different cell types.

#### 3.1.1. EV Cellular Uptake

The uptake of EVs is a complex process that involves a combination of different pathways [51], such as caveolae-dependent endocytosis, clathrin-dependent endocytosis, phagocytosis, pinocytosis, receptor-mediated endocytosis and fusion with the plasma membrane [52,53,54,55,56]. This variety of mechanisms provide EVs with some significant advantages in comparison to other synthetic drug delivery systems with respect to their mode of interaction with host cells and their ability to transfer therapeutic molecules [57]. There is still no consensus on which of the uptake mechanisms is more important; however, available evidence shows that internalization is an active process that is significantly decreased at 4 °C or after blocking protein interactions using specific antibodies. The high degree of EV heterogenicity depending on their cellular origin contributes significantly to their uptake and available evidence suggests that this can be selectively enhanced by choosing specific cell models [13,58].

#### 3.1.2. Cell Type-Dependent Uptake

There is increasing evidence that the specificity of EV uptake depends on their cellular origin [59,60,61]. For instance, EVs from mesenchymal stem cells are taken up more efficiently by their own cells when compared to other immune cells [62]; neuroblastoma EVs are endocytosed preferentially by glial cells [63]; ovarian cancer EV cells also showed increased uptake by tumor cells when compared to epithelial cell-derived EVs [64]. A recent study compared the uptake of multiple-source EVs with their corresponding cells and showed that melanoma EVs were taken up more efficiently by melanoma cells rather than immune cells, fibroblast and endothelial cells [13]. Interestingly, the uptake of colon adenocarcinoma EVs (by melanoma cells) was also higher than non-tumor EVs, suggesting that cancer-specific molecules may play an important role in promoting the uptake of these vesicles by tumor cells. Although there is still not enough evidence for the cell type-specific uptake in vivo models, it could potentially represent an effective strategy to direct drugs towards specific cells and reduce the therapy-associated side effects. The mechanisms involved in the preferential uptake of EVs are also not totally understood; however, there is evidence that tetraspanins and integrins play an important role in target selection. For example, Tspan8 is important for the uptake and targeting towards endothelial, kidney and pancreas cells, while CD151 is more important for lung and lymph node cell uptake [65]. On the other hand, EV expression of other surface proteins, such as ICAMs, has been implicated in enhanced uptake by immune cells [66]. Alternatively, surface heparan sulfate proteoglycans have been shown to play an important role in the uptake by cancer cells [67]. We summarized some examples of the evidence for enhanced uptake of specific EVs by different cells in Table 1. As we discuss in the following sections, improving tracking techniques will be essential to corroborate this hypothesis and exploit EV potential to target specific cells.

#### 3.1.3. EV Biodistribution

EVs also have natural advantages relating to their biodistribution, such as their reduced aggregation potential and the ability to avoid clearance by the reticuloendothelial system (RES) by presenting antiphagocytic markers [42]. Their small size favors their accumulation in highly vascularized tissues with low lymphatic drainage such as tumors, due to the enhanced permeability and retention (EPR) effect, which can be used as a strategy to increase targeting towards tumors. The distribution of the EVs depends on different factors, such as the administration route, cellular origin, concentration and time. In circulation, most EVs are captured by the mononuclear phagocytic system and delivered primarily to the liver, followed by spleen, lungs and the gastrointestinal tract [37]. This pattern of accumulation is similar to that of other NPs, such as synthetic liposomes, suggesting that the size also plays an important role in the uptake of EVs. The administration route is important in determining the biodistribution and accumulation of the EVs. For instance, intraperitoneal administration results in increased accumulation in the pancreas, while subcutaneous injection can result in increased accumulation in the GI tract [37]. Intravenous injection is the most studied administration route and usually results in increased delivery to the liver and spleen and reduced accumulation in pancreas [37]. Intratumoral administration has also been used to facilitate tumor accumulation, which is elevated in comparison with other NPs, such as liposomes [57]. Intranasal administration is also interesting as an effective strategy to promote EV-delivery to the brain [3]. Oral administration represents the overall preferred route of administration in patients, but is one of the most challenging options due to the multiple barriers of the digestive system. Curiously, EVs isolated from plants, milk and intestinal epithelial cells have an unique potential to reach the circulation after oral administration, which makes them interesting carriers to improve the bioavailability of drugs [68]. There is currently limited information available concerning the clearance of the extracellular vesicles due to technical difficulties related to their rapid uptake and the non-specific transfer of the fluorescent dyes to extracellular proteins [69]. Recently, some authors have shown that EVs have a short half-life in circulation and their blood concentration is determined by a complex balance between secretion into the bloodstream and rapid blood clearance (fast uptake by multiple cells) [70].

#### 3.1.4. EV Organ Tropism

The cell of origin of the EVs also has an important effect on their distribution and tropism to specific organs. For example, EVs from dendritic cells show increased accumulation in spleen, while melanoma EVs are more likely to accumulate in lungs [37]. Interestingly, some EVs show tropism towards organs related to their cellular origin (Table 1). Endothelial EVs from the brain can accumulate in the cerebral tissue [71], while melanoma EVs target preferentially melanoma metastases [13]. MSC-EVs showed increased accumulation in kidneys of mice with acute kidney injury, suggesting that the presence of a particular disease also affects their biodistribution [72]. Although the specific targeting of EVs is still not totally understood, there have been some advances in discovering important molecules involved in the process. For instance, integrins are important cell adhesion molecules involved in organ-specific metastasis of tumor cells [73,74]. Interestingly, as shown in Table 1, different adhesion molecules have an important impact in the organ tropism (e.g., lung, liver and pancreas). For example, the presence of α6β4 and α6β1 is important in EV tropism to the lungs, while expression of αvβ5 promotes accumulation in the liver [74,75]. There is also evidence that CD47 expression in EVs is important for avoiding phagocytosis, increasing time in circulation and uptake by micropinocytosis [42]. It is important to mention that a controversy exists concerning the distribution of EVs from different cell sources and, contrary to what we mentioned above, some articles reported a similar distribution of EVs isolated from different cell models [76,77]. Therefore, more information and better experimental approaches are needed to test this hypothesis

### 3.2. Therapeutic Properties

#### 3.2.1. Antitumoral Effects

EVs are involved in various processes related to the development and progression of cancer. For this reason, they are of great interest as possible diagnostic markers and potential candidates for directed therapy against malignant cells. Their natural potential to accumulate in specific tissues (as described above) makes them ideal systems to improve the delivery towards tumors located in zones that are difficult to access, such as the brain or hypoxic tumor sites. As other NPs, EVs benefit from the enhanced permeability and retention (EPR) effect, which allows them to passively accumulate in the tumors. As we mentioned before, there is evidence that tumor EVs can actively and preferentially target other tumor cells, which is an interesting approach for drug delivery purposes. Moreover, EVs have been used for antigen presentation strategies (vaccinations strategies) and to promote inflammation or cytotoxic effects in tumor cells (inflammation strategies) as we will summarize below and in Figure 3.

Antigen presentation strategies. Tumor cell-derived EVs (TEVs) have been shown to carry cancer antigens that promote antitumor effects by stimulating dendritic cells (DCs), which is an interesting strategy to enhance the immune responses against cancer cells [81]. For example, patient TEVs have been shown in clinical trials to produce specific antitumor immune cell responses [82]. As tumor antigens on TEVs are recognized by immune cells, once sensitized, they can selectively target and destroy malignant cells. Therefore, this strategy holds great promise to be used in the future as a way to develop cancer vaccines. Other authors have shown that EVs from malignant cells can also activate T lymphocytes and NK cells to facilitate the transfer of antigens to DCs. The antigen-activated immune cells can be administrated to the patient/animal (instead of tumor EVs) which subsequently increase the immune response. In this way, the side effects related to the administration of malignant EVs can be avoided [8,83,84]. Another related strategy involves using immune cell-derived EVs, as they can express specific markers, such as MHC molecules or CD3 (important for immune effects) and can be previously activated to generate a specific antitumoral effect. An example of such an approach which has reached clinical trials is using tumor cell peptides to activate DCs and then isolating EVs from activated-DCs to promote an immune response against cancer [19,83,85,86].

Inflammation strategies. As natural killers, NK cells are provided with several lytic proteins that are used to destroy pathogens. NK-EVs not only possess specific NK markers but also contain proteins, such as FasL, perforins, granzymes and granulysin, which promotes cytotoxic activity and have been effectively used to kill cancer cells [87]. The utilization of macrophage-derived EVs has also emerged as a novel therapeutic approach, because of their ability to modulate the tumor microenvironment and increase the sensitivity of tumor cells to drugs. While tumor-associated macrophages have been shown to promote tumor progression, angiogenesis, immunosuppression and metastasis, a study by Cheng et al. describes that their EVs actually promote inflammation-related responses, which can enhance immunotherapy effects against cancer cells [88]. Moreover, other studies suggest that by using EVs from macrophages polarized towards an M1-like phenotype (which has antitumoral features), it is possible to decrease the proliferation of malignant cells, enhance immune responses and increase the effectiveness of drugs [89].

#### 3.2.2. Modulation of Inflammation

While acute inflammation plays a positive role in our defense systems against pathogens, prolonged inflammation can lead to the development of several chronic diseases, such as arthritis, cancer, Alzheimer, diabetes and cardiovascular diseases. Extracellular vesicles play an important role as mediators of pro-inflammatory or anti-inflammatory responses which can be exploited to treat chronic diseases (Figure 3). For instance, T cell-derived EVs inhibit production of the pro-inflammatory cytokines IL-1β and TNF [90], while neutrophil-derived EVs contain anti-inflammatory proteins that inhibit leukocyte/endothelium interactions [91]. Platelets also participate in important anti-inflammatory effects by driving the differentiation status of macrophages, dendritic cells and T cells towards less-reactive states [92,93]. Endothelial EVs promote anti-inflammatory effects by reducing ICAM-1 expression [94] and together with platelets have been attributed a protective role in sepsis [95]. MSC-derived EVs have been reported to reduce T lymphocyte proliferation, as well as the percentage of CD4+ and CD8+ T cell subsets [96]. Macrophage EVs also play an important role in immune surveillance after exposure to mycobacterial infections as they transport bacterial components that can induce pro-inflammatory responses [18,97]. Interestingly, some authors proposed that EVs could also be used to modulate aging speed by regulating age-related pro-inflammatory status [98,99]. Evidence suggest that EVs are involved in a crosstalk between telomere dysfunction and inflammation which contributes to aging-related disorders [100]. As EV miRNA profiles changes during aging, the use of EV from young donors could be an interesting approach to increase cell longevity [98,99,101].

#### 3.2.3. Cell Regeneration

Mesenchymal stem cells (MSCs) have revealed great potential in tissue regeneration. Their administration, however, may lead to serious side effects, such as differentiation into undesirable tissues, or inducing unfavorable immune responses. Some authors have indicated that MSCs induce such effects by paracrine signaling rather than direct cell-to-cell interactions. This has led to the hypothesis that their therapeutic effect may be mediated by EVs [102]. Indeed, MSC-EVs may represent an interesting approach to avoid the side effects of whole cell therapies. For example, MSC-EVs have been shown to accelerate skeletal muscle regeneration and promote hepatic regeneration after liver injury [103,104]. By equilibrating peripheral immune responses, MSC-EVs can also improve neuronal regeneration and prevent post-ischemic immunosuppression after intravenous administration [102]. Other EVs are also provided with interesting features for cell regeneration, such as endothelial EVs which can increase the number and differentiation state of human endothelial progenitor cells [105]. Schwann cell-derived EVs (SC-EVs) are also interesting, as they have been shown to enhance axonal regeneration and neuronal survival after the application of damaging stimuli to neurons [106]. Interestingly, these effects were specific for SC-EVs and mediated by the inhibition of the GTPase Rho, involved in axon retraction. These examples suggest that EVs from different sources may be useful in promoting tissue regeneration after damage (Figure 3).

#### 3.2.4. Cardiovascular Diseases

EVs are implicated in cardiovascular homeostasis, where they play an important role in preventing damage during stress conditions, which makes them interesting candidates for cardioprotective applications. Evidence suggest that EVs from cardiac progenitor cells can inhibit cardiomyocyte apoptosis and improve cardiac function after myocardial infarction [107]. MSC-EVs have also shown great potential in preventing cardiac tissue damage. Some remarkable findings are their potential to increase myocardial viability and reduce ischemia/reperfusion damage to the myocardia [108] and kidney [109]. The mechanism of cardioprotection is suggested to be linked to a decrease in oxidative stress and activation of the PI3K/Akt pathway, which increases viability and promotes remodeling of the myocardia [110]. Interestingly, such beneficial functions also apply to renal stress conditions, as MSC-EVs can protect against acute tubular injury [111] and reduce damage of ischemia/reperfusion-induced acute renal failure [112].

### 3.3. Modulating the Targeting of EVs

EV targeting can be modified using different approaches, such as changing the isolation method, subjecting the cells to stress prior to isolation or changing the administration route. Targeting of EVs can be further changed by introducing structural modifications to enhance different properties, such as stability and homing towards specific cells. Combining the natural properties of EVs with pre-cargo and post-cargo strategies can enhance the potential of EVs for therapeutic applications, targeting and imaging.

#### 3.3.1. Isolation Conditions

In addition to the cell source, it is also important to consider the isolation conditions as they can have an important impact on the uptake or biodistribution of the EVs. For example, EVs isolated by ultrafiltration and ultracentrifugation yielded different purity, and protein/vesicle ratios which resulted in different patterns of in vivo accumulation in lungs [113]. Antibody-based isolation methods can also impact on EV targeting as they can select for specific vesicle subpopulations (e.g., CD63+ vesicles), which may lead to alterations in the distribution pattern. Subjecting the cells to different stimuli before isolation has also been shown to have important effects on EV targeting. For instance, the uptake by hypoxic tumor cells can be enhanced by using EV-derived from cells subjected to hypoxia prior to isolation [79]. Similar effects have also been observed when using EVs from cells exposed to other stress conditions, such as radiation, thermal stress or changing the pH of the medium [114,115]. The mechanisms that explain the increased targeting after stress are not totally understood; however, there is evidence that the cargo content of EVs can change under these conditions [116]. For example, after pH stress VEGF accumulates inside EVs in its bioactive form, which favors the stimulation of recipient cells [114]. The lipid composition is also modified under such conditions, changing EV rigidity and increasing fusion efficiency [115]. Finally, the protein HSP90, which is involved in adhesion and migration, is also increased after heat-shock stress [116,117]. Subjecting cells to stress may not only improve targeting but also their therapeutic properties. ER stress, for example, can promote the release of pro-inflammatory EVs that stimulate macrophage chemotaxis by interfering with sphingosine-1-phosphate signaling [118].

#### 3.3.2. Engineering Targeting

As with other drug delivery systems, the natural targeting of EVs can be manipulated to improve their uptake and biodistribution. The surface modification or addition of a targeting ligand can be achieved by direct strategies involving the manipulation of isolated EVs or indirect strategies involving the modification of cells before EV isolation. The most common strategies are summarized in Figure 4 and discussed below. We also provide some examples of multiple engineered targeting strategies in Table 2.

Covalent linkage: In this case, the desired targeting molecule is attached to the EV surface by a chemical reaction. The amine/carboxylic groups present in the membrane of the EVs can be functionalized using the EDC/NHS reaction giving rise to stable covalent amide bonds. This strategy can be combined with click chemistry, which permits incorporating complex ligands in a fast reaction with high specificity [12,14,119].

Electrostatic interaction: This strategy refers to the utilization of positively charged molecules that can be attached to the EVs via electrostatic interactions. This is possible because the EV surface is negatively charged, so this simple method does not require any chemical reactions [120].

Direct lipid linkage: This method involves attaching the targeting ligand to a lipid or lipid-like molecule, which will spontaneously insert into the vesicle surface exposing the targeting compound. An example here is the coupling of a cardiac homing peptide to the neutral phospholipid dioleoylphosphatidylethanolamine (DOPE). This is achieved by conjugating DOPE-NHS with the desired peptide for targeted heart therapy [121].

Indirect lipid linkage: An alternative to the previous method is to incubate the cells with a lipid complex conjugated to polyethylenglycol (PEG) which is incorporated into the membrane and then released on the surface of EVs. Pegylation of EVs is an excellent way to enhance stability, circulation time and allows further functionalization with other molecules to enhance targeting [15]. For example, a DSPE–PEG–RGD complex was incubated with K562 cells for membrane inclusion. The complex was later found on EVs which had increased targeting towards blood vessels [122].

Cell transfection: This is a more complex process, as it involves engineering cells to secrete EVs including the desired targeting molecule. This process is achieved by transfecting genetic material into cells which is then stably or transiently incorporated in the cells to produce the target protein. The effectivity of this strategy was demonstrated by Tian et al. 2014, who engineered cells to express the EV membrane protein Lamp2b fused to an αv integrin-specific peptide for targeted antitumor therapy [123]. Similar approaches have been studied to target EVs towards chronic leukemia cells [124], brain [28] and cardiac tissue [9].

#### 3.3.3. Tracking Studies of EVs

The use of EVs likely represents a key approach to selectively target organs; however, many technological limitations exist that make it difficult to analyze the effects of these vesicles in vivo. In order to exploit the targeting properties of the EVs for multidisciplinary applications, it is necessary to be able to track their uptake and distribution throughout the organism. Currently, several imaging techniques are available in the clinic for diagnostic applications, such as computed tomography (CT), magnetic resonance imaging (MRI), X-ray, ultrasound, positron emission tomography (PET) and single-photon emission computed tomography (SPECT). All of these techniques could potentially be used to image EVs with high sensitivity and endow them simultaneously with therapeutic properties to allow theranostic applications as we discuss below and in Figure 5.

Fluorescence is the most common way of EV tracking because of its high versatility. EVs can be labelled simply by incubation with a great variety of lipophilic fluorescent markers that combine different excitation/emission wavelengths. It has been suggested that the lipophilic marker is quickly stabilized in the membrane of the extracellular vesicles. As tissue transparency is attained only using wavelengths above 780 nm (therapeutic window), near-infrared dyes have become quite popular and are currently the most used for in vivo imaging of EVs. Although tracking fluorescence is a widely used technique for imaging, it is still not a reliable technique for clinical use. The stability and performance of fluorophores are limited by several effects such as scattering, tissue autofluorescence and photobleaching. In addition, they can be easily degraded by oxidation reactions in the organism. Recently, it has been reported that lipophilic fluorescent dyes can be transferred from EVs to other extracellular components, leading to unspecific labelling [69]. Moreover, dyes can spontaneously form micelles which also stain cells, limiting the conclusions that can be drawn concerning uptake and distribution of EVs [69,127]. Naturally expressing fluorescent proteins such as GFP/RFP by transfecting cells would result in a more stable labelling. However, these are still limited by their fluorescence in the visible light spectrum which makes them poorly reliable for in vivo analysis [128]. Therefore, novel approaches have been developed to improve EV imaging. As an alternative to the commonly used fluorescence probes, Zhang et al. 2018 proposed indirectly labelling EVs by covalently conjugating membrane phospholipids with fluorescent markers, which permits reducing the non-specific extracellular labelling [129]. By taking advantage of the sulfhydryl groups on the EV surface, maleimide-conjugated Alexa-fluor dyes were covalently added to the EV surface by simple incubation [130]. Protein-based labelling of EVs has also been achieved by using luciferase reporters on cells to produce bioluminescent proteins that are later transferred to the EVs which permit stable real time monitoring [131,132].

Gold nanoparticles are another very interesting class of nanomaterials that can be used for EV tracking because of their high tunability, biocompatibility and unique optical properties related to their surface plasmon resonance [133,134]. These properties allow AuNPs to be used for enhancing CT imaging, photoacoustic imaging and photothermal therapy, which is useful for simultaneous imaging and tumor ablation in cancer therapy. These NPs have been effectively incorporated into EVs and used for therapeutic purposes, to analyze the EV sorting pathways in cells, as well as for CT analysis [3,135]. Betzer et al. 2017 incorporated gold nanoparticles into EVs from MSCs and used them for in vivo neuroimaging by computer tomography and ex vivo analysis by inductively coupled plasma spectrometry (ICP), which allows quantitative analysis of EV distribution [3]. As direct gold labelling techniques may affect the distribution, we developed a double labelling method to incorporate AuNPs indirectly into EVs by incubating them with cells and combining this with direct fluorescent labelling of the EVs to permit analyzing EV biodistribution by neutron activation analysis, NIR fluorescence, CT imaging and gold-enhanced microscopy imaging. We showed that the indirect gold labelling strategy did not affect biodistribution and were able to identify the presence of AuNP-labelled melanoma EVs in small metastatic foci in the animal lungs [13].

Superparamagnetic iron oxide nanoparticles (SPIONs) are also interesting systems for imaging as they can be used for magnetic resonance imaging. They have the additional advantage that they can be used to facilitate EV isolation using magnets [40,136] and for simultaneous therapy using a magnetic field [40]. These NPs have also been effectively incorporated into EVs and used for in vivo imaging by magnetic particle imaging [79] and MRI [2,43,137]. Other authors have encapsulated quantum dots (QDs) into EVs. These strongly fluorescent materials could be useful for variety of fluorescence analysis techniques, because of their enhanced stability and the possibility of using them simultaneously for photodynamic therapy [138].

Combining multiple imaging compounds may represent a workable strategy for multimodal imaging in combination with therapy. Rehman et al. 2018 developed an interesting method to indirectly load gold and Fe nanoclusters into EVs. The authors pre-incubated cells with the salts HAuCl4 and FeCl2 to produce the required nanoclusters in cells and then release them as EV cargos [139]. Although the authors did not provide evidence that the EVs retained their biodistribution after the loading, this may represent an interesting strategy to load EVs with multiple NPs for theranostic applications.

## 4. The potential Risks and Limitations Associated with the Use of EVs

### 4.1. Therapeutic Risks

EVs have great potential for the treatment and prevention of several diseases. However, it is important to consider that there are multiple limitations associated with their utilization and little information exists concerning the risk after prolonged exposure to these natural signaling vesicles. Several authors have reported studies that reveal the risk of EVs in cancer, particularly of tumor-derived EVs (TEVs), which have been shown to facilitate drug resistance, tumor progression and metastasis [20,140,141]. Some important examples include the role of melanoma EVs in facilitating metastasis by supporting tumor growth, preparing the metastatic niche, inducing vascular leakiness and reprograming of bone marrow cells [141]. Further, TEVs have been reported to promote tumor growth by suppressing NK cell function [142], inducing lymphocyte apoptosis [143] and driving epithelial mesenchymal transition [140]. The presence of a variety of signaling molecules in the EVs is often linked to these effects. For example, the presence of CAV-1 in EVs from breast metastatic cells is associated with enhanced migration and invasion of recipient cells [144] and the EV transfer of MET with melanoma metastasis to the bone marrow [141]. Recent studies also point towards negatives effects in the cardiovascular system. For example, cardiac fibroblast-derived EVs have been shown to activate the angiotensin II receptor type I pathway, which is involved in hypertrophy and heart failure [145]. Moreover, neutrophil EVs have been shown to contribute to vascular inflammation and atherogenesis [146]. There is currently limited information about approaches focusing on modifying EVs to ameliorate these problems without affecting their therapeutic potential. For example, an interesting challenge ahead will be to develop a strategy that permits removing the malignant cargos from TEVs while maintaining their targeting and immunomodulatory properties.

### 4.2. Technical Limitations

In addition to their therapeutic risk, there are several technical challenges that need to be addressed for effective commercialization of EVs as biological drugs. Considering their heterogeneity, the preparation of EVs from cell cultures or blood needs to be standardized to obtain reproducible batches with similar biological activity. Good manufacturing guidelines should be developed to obtain products with similar quality. Producing large amounts of EVs is essential for worldwide marketing. Therefore, scaling-up strategies should be developed by using novel isolation methods or improving the limitations of the current ones (e.g., low yield and poor sorting of EV subpopulations). Some authors suggest that for MSCs, a total of 500 million cells will be needed to obtain enough EVs for a given therapeutic intervention. This value was extrapolated from experiments in mice considering that 80 µg is approximately the amount of exosomes released from 2 million MSCs in 48 h [147,148]. Achieving this will require improving current manufacturing strategies, such as bioreactor culture platforms, as well as concentration and separation technologies [148]. The stability of the EVs is also an issue as the vesicle content of proteins or RNA can be altered by changes in temperature. Currently, −80 °C is the most accepted storage condition to preserve EVs for longer periods of time. However, multiple freeze/thawing cycles of the samples can also alter their vesicular structure and their biological activity [149]. Therefore novel stabilization strategies, such as using buffers or polymers, should be developed as a way to overcome these issues. Finally, the characterization techniques should also be improved in order to determine the molecular composition of each type of EV, standardize their quality and determine the batch-to-batch variability between samples [150]. These procedures will be of particular importance for clinical applications of EVs from external sources (e.g., plants, milk or patient EVs), which are mostly limited by their variability and will greatly benefit from quality standardization. In the future, we may be able to profile the therapeutic properties of EVs and generate easily-scalable synthetic systems that contain specific EV molecules for personalized medicine.

## 5. Summary and Future Perspectives

EVs have shown great potential for the development of multiple drug delivery and vaccination strategies. As natural modulators of cellular communication, EVs provide a unique source of biological information which will depend on their cellular origin and can be exploited for therapeutic or diagnostic approaches. EVs can provide multiple benefits in comparison with other drug delivery systems. Here, we reviewed some of their natural advantages, which include their organ tropism, cardioprotective effects, and antitumoral properties, as well as their ability to modulate inflammation and promote tissue regeneration. These effects could be useful for more personalized medicines in order to increase the effectiveness or reduce the side effects of drugs, for instance by modulating the tumor microenvironment to overcome drug resistance and reducing the cardiotoxicity of chemotherapy. The evidence of cell type-dependent uptake and distribution holds great promise for targeted therapy. Exploiting these properties obliviates the need for functionalization or chemical modifications, thereby facilitating translation to the clinic. Detecting and analyzing EV distribution are important in order to understand their trafficking inside our body and the mechanisms involved as well as to uncover new adhesion molecules that determine their homing selectivity. This information will also help to develop novel synthetic delivery systems with similar homing properties and fewer side effects than EVs that can be produced on an industrial scale. Multimodal imaging of EVs may be achieved by using different types of nanoparticles and fluorophores which, when combined with the targeting abilities of these vesicles, could lead to the development of powerful tools in theranostics. As novel technological approaches arise, we may be able to determine and control the risk associated with EVs after prolonged exposure. Standardization of the isolation of different EV subtypes and upscaling their preparation will be essential to exploit the potential of these very heterogeneous membrane-based structures. In the future, EVs loaded with contrast agents could be used for early diagnosis of diseases, such as cancer, by imaging techniques such as CT and MRI. Further, they may serve as vectors for the selective delivery of drugs to specific parts of the body, thereby reducing side effects and improving patient outcome.

## Figures and Tables

**Figure 1 pharmaceutics-12-01022-f001:**
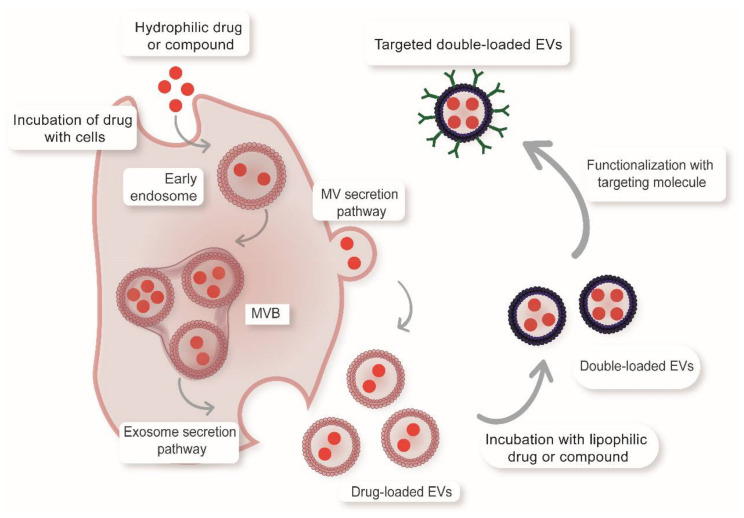
Simultaneous incorporation of hydrophilic drugs, lipophilic compounds and targeting agents into EVs. Pre-cargo and post-cargo strategies can be combined to simultaneously load EVs with multiple therapeutic agents. Loading hydrophilic molecules can be achieved without membrane surface disruption by incubating cells with the cargo molecules. After the cargo is incorporated by the cells, it may follow the endocytic pathway and be released inside exosomes or be directly liberated from the cell surface inside MVs. The resulting EVs carrying the therapeutic cargo can additionally be loaded with small lipophilic molecules, such as a fluorescent dyes or drugs by direct incubation strategies. Finally, the resulting double-loaded EVs can be externally modified with targeting agents (e.g., peptides, antibodies, drugs) by chemical approaches.

**Figure 2 pharmaceutics-12-01022-f002:**
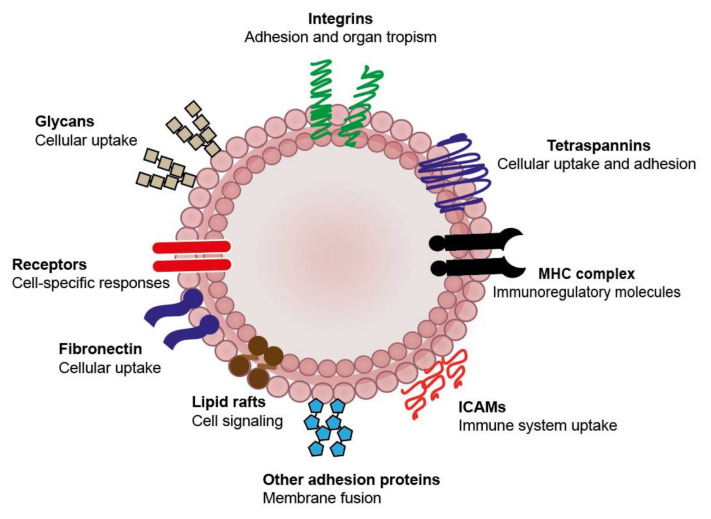
Naturally occurring features of EVs for targeting applications. EVs contain several surface molecules than will determine their systemic distribution and targeting to specific cell populations. Tetraspannins, glycans, fibronectin and intercellular adhesion molecule (ICAM) composition modulate their uptake, while integrins play an important role in defining their accumulation in specific organs. Immunoregulatory molecules influence their recognition by immune cells which can have an impact on their distribution and uptake. Lipids, receptors and cell-specific proteins can also modulate the responses of target cells by activating multiple cell signaling pathways.

**Figure 3 pharmaceutics-12-01022-f003:**
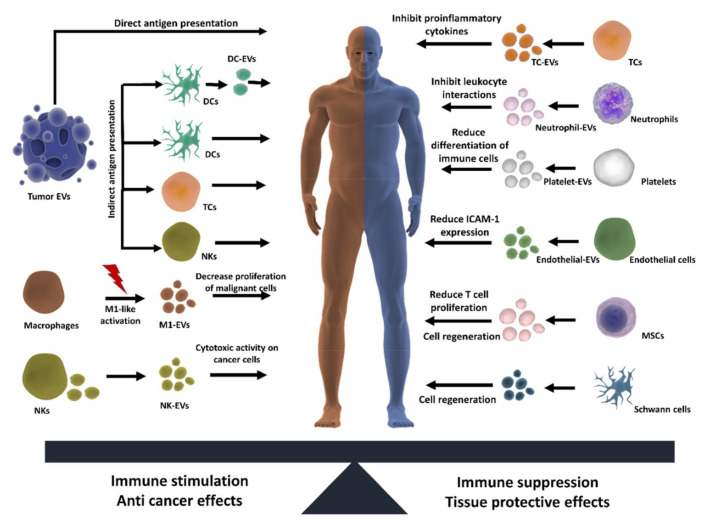
Using the natural features of EVs for therapy and tissue repair. EVs play a key role in modulating cellular responses by enhancing or suppressing immune responses in target cells. These properties can be exploited for therapeutic purpose by isolating EVs with the desired characteristics (e.g., antitumoral activity, cell regeneration or cardioprotection). Combining these effects together to modulate the target microenvironment is an interesting approach to reduce the side effects of drugs, improve efficacy and generate more personalized medicines. DCs: dendritic cells; NKs: natural killer cells; TCs: T cells; MSCs: mesenchymal stem cells.

**Figure 4 pharmaceutics-12-01022-f004:**
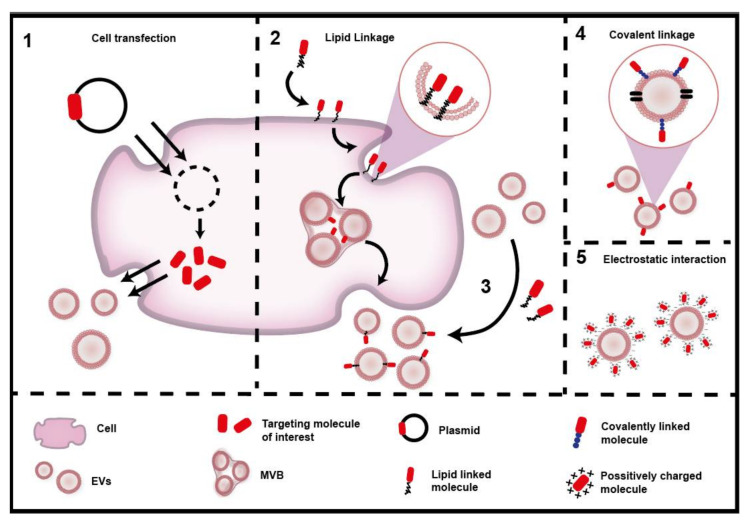
Engineering targeting strategies. EV targeting can be manipulated by adding targeting ligands using different strategies. (1) Transfection of the cell with a plasmid to generate the target protein within the cell and thereby facilitate secretion in EVs. (2) Modification of the targeting molecule with a lipid that can be anchored to the membrane of the cells and then indirectly secreted on EVs or (3) by directly adding lipid-linked molecules to the surface of EVs. (4) Covalently linking the targeting agent into the surface of EVs. (5) Using a cationic compound to promote electrostatic interactions with the EVs.

**Figure 5 pharmaceutics-12-01022-f005:**
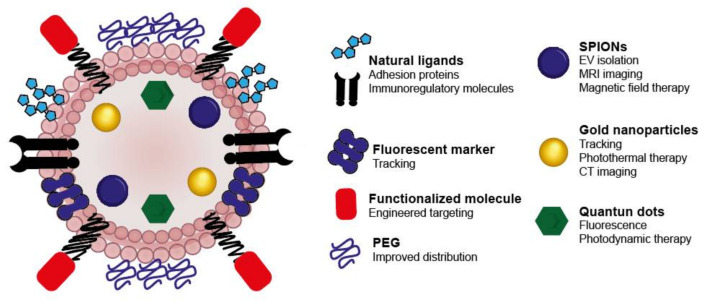
Theranostic EV. Combining the natural properties of EVs with drug loading and targeting strategies leads to multiple therapeutic benefits, such as enhanced uptake/targeting, prolonged circulation time, immunomodulation, tracking and therapy. The utilization of NPs is interesting, as they can add multiple features simultaneously (e.g., tracking and therapy) reducing the requirement of multiple drugs. SPION: superparamagnetic iron oxide nanoparticles; CT: computed tomography; PEG: polyethylenglycol.

**Table 1 pharmaceutics-12-01022-t001:** Examples of natural EV targeting to specific cells.

Isolated from/Enriched in	Targeting to	Reference
Human placental MSCs	MSCs	[62]
Neuroblastoma N2a cells	Glial cells	[63]
Brain endotelia bEND.3 cells	Brain	[71]
Bone marrow DCs	Spleen	[37]
Ovarian cancer SKOV3 cells	SKOV3 cells	[64]
Melanoma B16F10 cells	B16F10 cells, lungs and pulmonary metastasis	[13,37]
Melanoma B16BL6 cells	B16BL6 cells	[60]
Bone marrow MSCs	kidneys on acute kidney injury model and M2 type macrophages on injured spinal cord	[72,78]
Fibroblast CD47+	Pancreatic cancer	[42]
Breast cancer MDA-MB-231 under hipoxia	Hypoxic MDA-MB-231 cells	[79]
Heparan sulfate proteoglicans on hepatic cell lines AML12 and MLP29	Hepatic (Huh7), fibroblastoid (M1) cells and osteoblast (U2-OS) cells	[67]
Tspan 8 expresion in pancreatic adenocarcinoma cells BSp73AS	Pancreas and lung	[65]
Fibronectin in microvascular endothelial cells MVECs	Oligodendrocyte precursor cells	[80]
ICAM on bone marrow DCs	Naïve T cells	[66]
αvβ5 expression on multiple cells (refeer to publication)	Liver tropism	[75]
α6β4 and α6β1 expression on multiple cells (refeer to publication)	Lung tropism	[75]

**Table 2 pharmaceutics-12-01022-t002:** Examples of EVs engineered strategies to improve targeting towards specific cells.

Method	Isolated from	Engineered with	Targeting to	Reference
Covalent linkage	Bone marrow MSCs	αvβ3 integrin targeting peptide c(RGDyK)	Ischemic brain	[12]
	Macrophages RAW 264.7	neuropilin-1-targeted peptide (RGERPPR)	Glioma	[14]
Direct lipid linkage	Macrophages RAW 264.7	Sigma receptor targeting ligand z(aminoethylanisamide, AA)	Pulmonary metastasis	[125]
	Cardiosphere-derived cells	cardiac homing peptide CSTSMLKAC (CHP)	Myocardial infarction	[121]
Indirect lipid linkage	Leukemia cells K562	αvβ3 integrin targeting peptide c(RGDyK)	Angiogenic blood vessel	[122]
	Umbilical vein endothelial cells (HUVECs)	Biotin	HepG2 tumors	[126]
Transfection	Immature mouse DCs	αvβ3 integrin targeting peptide (CRGDKGPDC) (iRGD)	MDA-MB-231 tumors	[123]
	Embryonic kidney cells (HEK293)	IL3 ligand	Chronic myelogenous leukemia (CML)	[124]
	Bone marrow DCs	αvβ3 integrin targeting peptide YTIWMPENPRPGTPCDIFTNSRGKRASNG (RVG)	Brain	[28]
	Embryonic kidney cells (HEK293)	cardiac-targeting peptide APWHLSSQYSRT (CTP)	Cardiac tissue	[9]
	Embryonic kidney cells (HEK293)	Epstein–Barr virus protein (gp350)	B cell tropism	[16]
Passive adsorption	MSCs	Spermine cationized pullulan	Liver tissue	[120]

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
