# Peer review of "Exploiting the Natural Properties of Extracellular Vesicles in Targeted Delivery towards Specific Cells and Tissues"

_pharmaceutics, 2020, doi:10.3390/pharmaceutics12111022_

Round 1

Reviewer 1 Report

The review by Lara et al focuses on the use of extracellular vesicles for theranostic applications. The review is very well written and the topics discussed quite pertinent for the community.

Below some specific comments:

1. please, check the abbreviations throughout the manuscript, such as for example, nanoparticles, extracellular vesicles and so on. Define one and use throughout the manuscript.

Line 37: "targeted more specifically to specific sites". Redundancy. I suggest to delete "more specifically".

Line 40: "exceptional physical and chemical characteristics". Please provide more information regarding the "exceptional".

Line 43 and elsewhere: refrain from using etc. Replace by, for example, between others or something similar.

Line 50: replace "to overcome this stage..." by "to overcome these barriers must cross others, such as, for example, the BBB".

Line 52: add "intended" before cellular location and add a "," after location. Also, replace "low vascularization" by "poor and/or leaky".

Line 58: "natural biological properties", such as???? Give examples of those properties.

Line 65: "is" should be "are".

Line 71: Add "or biological fluids" after "EV-producing cells".

Line 80-84: I suggest to change the order. I would start by "Evs can be found in different biological fluids, such as.... and are capable of transporting cytosolic and membrane proteins..."

Line 105 and 106: delete sentence starting with "Evs can be isolated.... It was already mentioned before.

Line 117: replace produce by create.

Line 121: causing "EV properties to be lost or altered/modified...

Line 126: "biodistribution and release of the drug". Give more information regarding how this is achieved.

Line 121-128: most of the information is based on Ref. 21 which is a review. Please provide more details and appropriate references.

Line 129: parental cells should be "parental EV-producing cells".

Line 130: "some gene product that is then" should be "gene of interest that will be included"

Line 144: "agents simultaneously". Please give some examples.

Line 146: "their exceptional natural properties". Be more concrete,

Line 169: "other immune cells". Provide a reference.

Line 195: what do you mean by cell model? Refers to the producing or recipient cell?

Line 203: "in circulation, most...". This sentence should be move above (after pancreas).

Line 218: "cell origin" should be "cell of origin".

Page 6: in the first column of the table you should always provide information regarding the cell of origin. For example, the ones that say "fibronectin, CD151 and so on" should mention the cell of origin.

Line 241: explain how the cell-like structure is important to access those zones.

Line 295: "tumour formation". This is still very controversial and if you want to discuss this you need to clearly highlight the evidence available and put it in context. Make sure that in this sentence you use proper references.

Line 299: correct the position of the reference.

Line 301: MSC-Evs: mention the route of administration.

Line 333: delete "Ej".

Line 337: add "have" before also.

Line 337: EV-derived cells? What is this?

Line 348-352: please discuss if amine groups are the only option.

Line 339: Describe how does stress improve targeting and how can we adequate the stressor to a specific application?

Line 353-355: what type of molecules can be added and give examples on how those can be used for targeting.

Line 358: define DOPE

Line 365: how sure are you they are "inside Evs"?

Line 368: "incorporated in the target cell". This is not always the case because we can have transient expression which is still sufficient to enrich Evs. Discuss this.

Line 373: please include in the table the peptide sequence.

Line 382: is it the technique itself that benefits from the targeting abilities or they can be used to monitor/improve EV biodistribution? Clarify.

Line 391: add "tissue" before autofluorescence.

Line 394: "...unspecific labelling". Provide a reference.

Line 398: replace low by poorly.

Line 408: "novel properties" such as? Give examples.

Line 423: "advantage than can" should be "advantage that they can be used"

Line 427: add "of" after variety

Line 455: please discuss possible solutions to ameliorate the above-mentioned problems. Sorting EV subpopulations or other?

Line 463: mention the limitations

Line 464: please provide the calculations used to reach the 500 million cells (does not need to be accurate but for the reader to have na idea depending on the dose adminitered, how often needs to de administered and so on...

Line 497: provide details regarding how to use EV for early diagnostic based on CT and MRI. This is a bit unclear to me since the Evs can´t be modified.

Author Response

Reviewer 1

Comments and Suggestions for Authors

The review by Lara et al focuses on the use of extracellular vesicles for theranostic applications. The review is very well written and the topics discussed quite pertinent for the community.

Below some specific comments:

please, check the abbreviations throughout the manuscript, such as for example, nanoparticles, extracellular vesicles and so on. Define one and use throughout the manuscript.

Some abbreviations were corrected (e.g. GNP and AuNPs) as requested

Line 37: "targeted more specifically to specific sites". Redundancy. I suggest to delete "more specifically".

This sentence was corrected as requested

Line 40: "exceptional physical and chemical characteristics". Please provide more information regarding the "exceptional".

This sentence was corrected as requested 

Line 43 and elsewhere: refrain from using etc. Replace by, for example, between others or something similar.

Corrected in all the manuscript

Line 50: replace "to overcome this stage..." by "to overcome these barriers must cross others, such as, for example, the BBB".

This sentence was corrected as requested

Line 52: add "intended" before cellular location and add a "," after location. Also, replace "low vascularization" by "poor and/or leaky".

This sentence was corrected as requested

Line 58: "natural biological properties", such as???? Give examples of those properties.

This sentence was modified in the manuscript 

Line 65: "is" should be "are".

This sentence was corrected as requested

Line 71: Add "or biological fluids" after "EV-producing cells".

This sentence was corrected as requested

Line 80-84: I suggest to change the order. I would start by "Evs can be found in different biological fluids, such as.... and are capable of transporting cytosolic and membrane proteins..."

The order was changed  as requested

Line 105 and 106: delete sentence starting with "Evs can be isolated.... It was already mentioned before.

This sentence was modified in the manuscript 

Line 117: replace produce by create.

This sentence was corrected as requested

Line 121: causing "EV properties to be lost or altered/modified...

This sentence was corrected as requested

Line 126: "biodistribution and release of the drug". Give more information regarding how this is achieved.

This sentence was corrected as requested

Line 121-128: most of the information is based on Ref. 21 which is a review. Please provide more details and appropriate references.

Appropriate references were added as requested

Line 129: parental cells should be "parental EV-producing cells".

This sentence was corrected as requested

Line 130: "some gene product that is then" should be "gene of interest that will be included"

This sentence was modified in the manuscript 

Line 144: "agents simultaneously". Please give some examples.

Examples are now provided as requested

Line 146: "their exceptional natural properties". Be more concrete,

This sentence was modified in the manuscript 

Line 169: "other immune cells". Provide a reference.

Appropriate references were added as requested

Line 195: what do you mean by cell model? Refers to the producing or recipient cell?

This sentence was modified in the manuscript 

Line 203: "in circulation, most...". This sentence should be move above (after pancreas).

The sentence was moved above before discussing the administration routes

Line 218: "cell origin" should be "cell of origin".

This sentence was corrected as requested

Page 6: in the first column of the table you should always provide information regarding the cell of origin. For example, the ones that say "fibronectin, CD151 and so on" should mention the cell of origin.

Cells of origin are provided as requested, with an exception for reference 74 in which multiple cells were used (+/- 28 cell lines) as indicated in the table

Line 241: explain how the cell-like structure is important to access those zones.

We corrected the sentence in the manuscript (section 3.2.1), the new sentence is:  Their natural potential to accumulate in specific tissues (as described above) makes them ideal systems to improve the delivery towards tumours located in zones that are difficult to access, such as the brain or hypoxic tumour sites

Line 295: "tumour formation". This is still very controversial and if you want to discuss this you need to clearly highlight the evidence available and put it in context. Make sure that in this sentence you use proper references.

We removed the words

Line 299: correct the position of the reference.

The reference position was corrected as requested

 Line 301: MSC-Evs: mention the route of administration.

The route of administration was provided as requested

Line 333: delete "Ej".

This sentence was corrected as requested

Line 337: add "have" before also.

This sentence was corrected as requested

Line 337: EV-derived cells? What is this?

This sentence was corrected and modified in the manuscript 

Line 348-352: please discuss if amine groups are the only option.

We included the carboxylic groups as other options

Line 339: Describe how does stress improve targeting and how can we adequate the stressor to a specific application?

We provided additional information, references and discussed this topic

Line 353-355: what type of molecules can be added and give examples on how those can be used for targeting.

Examples of molecules and their targeting are summarized in table 2

Line 358: define DOPE

DOPE was defined in the manuscript

Line 365: how sure are you they are "inside Evs"?

Corrected in manuscript:  “ The complex was later found on EVs which had increased targeting towards blood vessels”

Line 368: "incorporated in the target cell". This is not always the case because we can have transient expression which is still sufficient to enrich Evs. Discuss this.

We corrected the sentence to mention that both stable or transient expression can be used, as requested.

Line 373: please include in the table the peptide sequence.

Peptide sequences were provided as requested.

Line 382: is it the technique itself that benefits from the targeting abilities or they can be used to monitor/improve EV biodistribution? Clarify.

We clarified this sentence in the manuscript

Line 391: add "tissue" before autofluorescence.

This sentence was corrected as requested

Line 394: "...unspecific labelling". Provide a reference.

References were provided as requested

Line 398: replace low by poorly.

Replaced as requested

Line 408: "novel properties" such as? Give examples.

This sentence was corrected in the manuscript

Line 423: "advantage than can" should be "advantage that they can be used"

This sentence was corrected as requested

Line 427: add "of" after variety

This sentence was corrected as requested

Line 455: please discuss possible solutions to ameliorate the above-mentioned problems. Sorting EV subpopulations or other?

Additional material was provided in the discussion as requested

Line 463: mention the limitations

Examples of the limitations were provided as requested

Line 464: please provide the calculations used to reach the 500 million cells (does not need to be accurate but for the reader to have na idea depending on the dose adminitered, how often needs to de administered and so on...

Details were provided as requested

Line 497: provide details regarding how to use EV for early diagnostic based on CT and MRI. This is a bit unclear to me since the Evs can´t be modified.

We clarified this by stating that EVs can be loaded with contrast agents for imaging purposes.

Reviewer 2

The manuscript by Pablo Lara offers a very deep and complete information regarding EVs manipulation towards a clinical use. Obtention, uptake, distribution and content are some of the addressed topics in the text.

I have enjoyed the lecture, although I want to note some tiny details:

Line 217: 3.1.3. EVs organ tropism. I guess it should be 3.1.4.

This was corrected in the manuscript as requested

Line 384: please change “exosome” for “EVs”.

This word was corrected as requested

Line 473: “determine the batch to batch variability between samples”. I do agree with authors, although I would appreciate a brief comment regarding the use of self EVs for individualized treatments. As an example, it has been mentioned that EVs from young donors could exert beneficial effects of aged recipients. What if EVs could be isolated from a young donor, stored and later on, defrozen and engineered according to that donor specific condition? Could authors briefly address this topic?

A discussion of this topic was added in section 3.2.2. (modulation of inflammation) and in 4.1. (technical limitations) as requested

Reviewer 2 Report

In this manuscript, Lara and colleagues review the application of extracellular vesicles (EVs) as target drug delivery system in several diseases including cancer treatment, immunosuppression, cell regeneration, and cardioprotection. Authors also discuss natural EVs or engineered EVs targets specific tissues and organs. Interestingly, the current strategies to enhance efficacy of targeting of cargo/drugs to specific organs. This extensive knowledge could enhance our understanding of roles of extracellular vesicles in drug delivery. however, the authors should be aware of few issues in the current manuscript.

Some concerns:

  • Proteins was missed in Line 112, as Proteins has been reported to incorporated into EVs for the therapeutic purpose.
  • Please discuss more about electroporation in the application of siRNA, as it was reported aggregation rather than electroporated inside of EVs.
  • In table 1, please rearrange the table into a more adhesive way, such as group three Melanoma B16F10 Cells study together; having another column to have a easy understanding about targeted cell types and tissues/organs; add processing cell types in Tespan8, FN, CD151, ICAM, integrins, and HSPG in table1, also please add some comments on those examples for readers.
  • Recommendation on “Passive adsorption”, clearly electrostatic interactions is only example, why not directly use “electrostatic interactions” as subtitle. It also be stated in Fig4 cartoon as “5 Electrostatic interaction”
  • In table 2, either based on “cell types” or “Methods/strategies” or even “tissues” to categorize the table, which is easy to follow.
  • In graphical abstract, add examples (proteins) in each category, which makes more meaningful, similar like in Fig2
  • In Fig1 legend, it has been discussed on the cargo might be incorporated into MV, while MV part did not show in the cartoon.
  • In Fig2, please define TCs
  • In Fig5, please define SPOINs in legend. Authors used yellow particles as gold nanparticles in Fig5, while in Fig1, yellow particle define as cargo in EVs, suggesting to change cargo color in Fig1

Author Response

Reviewer 2

In this manuscript, Lara and colleagues review the application of extracellular vesicles (EVs) as target drug delivery system in several diseases including cancer treatment, immunosuppression, cell regeneration, and cardioprotection. Authors also discuss natural EVs or engineered EVs targets specific tissues and organs. Interestingly, the current strategies to enhance efficacy of targeting of cargo/drugs to specific organs. This extensive knowledge could enhance our understanding of roles of extracellular vesicles in drug delivery. however, the authors should be aware of few issues in the current manuscript.

Some concerns:

1.Proteins was missed in Line 112, as Proteins has been reported to incorporated into EVs for the therapeutic purpose.

Proteins were added as requested

2.Please discuss more about electroporation in the application of siRNA, as it was reported aggregation rather than electroporated inside of EVs.

Discussion was added as requested

3.In table 1, please rearrange the table into a more adhesive way, such as group three Melanoma B16F10 Cells study together; having another column to have a easy understanding about targeted cell types and tissues/organs; add processing cell types in Tespan8, FN, CD151, ICAM, integrins, and HSPG in table1, also please add some comments on those examples for readers.

Melanoma cells were grouped as requested

Processing cells lines were added as requested

Additional references to the table were added for the readers as requested

4.Recommendation on “Passive adsorption”, clearly electrostatic interactions is only example, why not directly use “electrostatic interactions” as subtitle. It also be stated in Fig4 cartoon as “5 Electrostatic interaction”

Text was modified as recommended

5.In table 2, either based on “cell types” or “Methods/strategies” or even “tissues” to categorize the table, which is easy to follow.

Table 2 was modified based on methods to allow easier follow us

6.In graphical abstract, add examples (proteins) in each category, which makes more meaningful, similar like in Fig2

The graphical abstract was modified as suggested

7.In Fig1 legend, it has been discussed on the cargo might be incorporated into MV, while MV part did not show in the cartoon.

MV secretion pathway is now added on figure 1

8.In Fig2, please define TCs

Abbreviations are now defined in the figure legend.

9.In Fig5, please define SPOINs in legend. Authors used yellow particles as gold nanparticles in Fig5, while in Fig1, yellow particle define as cargo in EVs, suggesting to change cargo color in Fig1

Abbreviations are now defined in the figure legend, the color of yellow particles was changed as requested

Reviewer 3 Report

The manuscript by Pablo Lara offers a very deep and complete information regarding EVs manipulation towards a clinical use. Obtention, uptake, distribution and content are some of the addressed topics in the text.

I have enjoyed the lecture, although I want to note some tiny details:

Line 217: 3.1.3. EVs organ tropism. I guess it should be 3.1.4.

Line 384: please change “exosome” for “EVs”.

Line 473: “determine the batch to batch variability between samples”. I do agree with authors, although I would appreciate a brief comment regarding the use of self EVs for individualized treatments. As an example, it has been mentioned that EVs from young donors could exert beneficial effects of aged recipients. What if EVs could be isolated from a young donor, stored and later on, defrozen and engineered according to that donor specific condition? Could authors briefly address this topic?

Author Response

Reviewer 3

The manuscript by Pablo Lara offers a very deep and complete information regarding EVs manipulation towards a clinical use. Obtention, uptake, distribution and content are some of the addressed topics in the text.

I have enjoyed the lecture, although I want to note some tiny details:

Line 217: 3.1.3. EVs organ tropism. I guess it should be 3.1.4.

This was corrected in the manuscript as requested

Line 384: please change “exosome” for “EVs”.

This word was corrected as requested

Line 473: “determine the batch to batch variability between samples”. I do agree with authors, although I would appreciate a brief comment regarding the use of self EVs for individualized treatments. As an example, it has been mentioned that EVs from young donors could exert beneficial effects of aged recipients. What if EVs could be isolated from a young donor, stored and later on, defrozen and engineered according to that donor specific condition? Could authors briefly address this topic?

A discussion of this topic was added in section 3.2.2. (modulation of inflammation) and in 4.1. (technical limitations) as requested

Reviewer 4 Report

In the manuscript by Lara et al., the authors reviewed the recent literature on the use of extracellular vesicles (EVs) as a diagnostic, prognostic, theranostic as well as a tool for drug delivery. After introductory paragraphs on EVs biology and biogenesis, they illustrate the techniques to modify EVs in order to incorporate or express on their surface drugs. The authors also depict also the ability of EVs of different origin to target different acceptor cells, to escape macrophage internalization and removal, to sustain EVs stability and half-life as well as to modulate pro- and anti-inflammatory responses. Finally, the authors underline risk connected to long term use of EVs as therapeutic tool and technical limitations that to date hinder the EVs application as pharmaceuticals.

The review is well conceived, organized and it appear complete and exhaustive in the description of not only the advantages but also the caveats in the use of EVs as diagnostic, prognostic, theranostic, therapeutic tools. For this reason, in my opinion it is suitable for publication in Pharmaceutics.

Just some typos have to be revised:

Line 83: Add comma after “.Importantly….”

Line 111: Add a point after “mononuclear phagocytic system [41]”

Line 264: Erase a space after “…. 88 ]”

Line 313: Add a space after “infarction” and before the ref. [99]

Line 340: specify what ER stands for in ER-stress.

Author Response

Reviewer 4
Line 83: Add comma after “.Importantly….”
This was corrected in the manuscript as requested
Line 111: Add a point after “mononuclear phagocytic system [41]”
This was corrected in the manuscript as requested
Line 264: Erase a space after “…. 88 ]”
This was corrected in the manuscript as requested
Line 313: Add a space after “infarction” and before the ref. [99]
This was corrected in the manuscript as requested
Line 340: specify what ER stands for in ER-stress.
Thank you, is ER in general.

Reviewer 5 Report

Manuscript is well written and can be published after minor revision.

Minor comments:

Line 80. It is known that EVs are released not exclusively by eukaryotic cells. The paper would benefit from pointing out that EVs releasing is evolutionary conservative process.

Line 82. EVs contain organelles as well. For more information see: doi:10.1038/nm.2736, doi:10.1523/JNEUROSCI.2429-08.2008, doi:10.3390/ijms15034142.

Line 176. The hypothesis that vesicles derived from tumor cells might be efficiently taken up by the same tumor cells has been verified recently (for instance doi: 10.1155/2018/7053623). The authors should mention these results.

Fig. 2 and Fig.5 are almost the same and might be merged in one. 

Author Response

Line 80. It is known that EVs are released not exclusively by eukaryotic cells. The paper would benefit from pointing out that EVs releasing is evolutionary conservative process.

The information is now provided as suggested by the reviewer

Line 82. EVs contain organelles as well. For more information see: doi:10.1038/nm.2736, doi:10.1523/JNEUROSCI.2429-08.2008, doi:10.3390/ijms15034142.

The information is now provided as suggested by the reviewer

Line 176. The hypothesis that vesicles derived from tumor cells might be efficiently taken up by the same tumor cells has been verified recently (for instance doi: 10.1155/2018/7053623). The authors should mention these results.

The requested publication was added as requested: “It is important to mention that a controversy exists concerning the distribution of EVs from different cell sources and, contrary to what we mentioned above, some articles reported similar distribution of EVs isolated from different cell models [73,74]. Therefore, more information and better experimental approaches are needed to test this hypothesis”

Fig. 2 and Fig.5 are almost the same and might be merged in one. 

Fig. 2 highlights the natural properties while fig. 5 summarizes the modifications that can be made in order to obtain theranostic EVs. In the graphical abstract, both figures were essentially fused to illustrate the overall potential of EVs
